# Evidence-Based Surgical Treatment Algorithm for Unstable Syndesmotic Injuries

**DOI:** 10.3390/jcm11020331

**Published:** 2022-01-10

**Authors:** Markus Regauer, Gordon Mackay, Owen Nelson, Wolfgang Böcker, Christian Ehrnthaller

**Affiliations:** 1Department of Orthopedics and Trauma Surgery, Musculoskeletal University Center Munich (MUM), University Hospital, LMU Munich, 81377 Munich, Germany; Wolfgang.Boecker@med.uni-muenchen.de (W.B.); Christian.Ehrnthaller@med.uni-muenchen.de (C.E.); 2Sportortho Rosenheim, 83022 Rosenheim, Germany; 3Health Sciences and Sport, University of Stirling, Stirling FK9 4LA, UK; gordonmmackay@gmail.com; 4Orthopedics and Sports Medicine, Waldo County General Hospital, Belfast, ME 04915, USA; oanelson@icloud.com

**Keywords:** syndesmosis, anterior inferior tibiofibular ligament, high ankle sprain, rotational instability, posterior malleolus, stabilization, anatomic repair, syndesmotic screw, suture-button, internal bracing, treatment algorithm

## Abstract

Background: Surgical treatment of unstable syndesmotic injuries is not trivial, and there are no generally accepted treatment guidelines. The most common controversies regarding surgical treatment are related to screw fixation versus dynamic fixation, the use of reduction clamps, open versus closed reduction, and the role of the posterior malleolus and of the anterior inferior tibiofibular ligament (AITFL). Our aim was to draw important conclusions from the pertinent literature concerning surgical treatment of unstable syndesmotic injuries, to transform these conclusions into surgical principles supported by the literature, and finally to fuse these principles into an evidence-based surgical treatment algorithm. Methods: PubMed, Embase, Google Scholar, The Cochrane Database of Systematic Reviews, and the reference lists of systematic reviews of relevant studies dealing with the surgical treatment of unstable syndesmotic injuries were searched independently by two reviewers using specific terms and limits. Surgical principles supported by the literature were fused into an evidence-based surgical treatment algorithm. Results: A total of 171 articles were included for further considerations. Among them, 47 articles concerned syndesmotic screw fixation and 41 flexible dynamic fixations of the syndesmosis. Twenty-five studies compared screw fixation with dynamic fixations, and seven out of these comparisons were randomized controlled trials. Nineteen articles addressed the posterior malleolus, 14 the role of the AITFL, and eight the use of reduction clamps. Anatomic reduction is crucial to prevent posttraumatic osteoarthritis. Therefore, flexible dynamic stabilization techniques should be preferred whenever possible. An unstable AITFL should be repaired and augmented, as it represents an important stabilizer of external rotation of the distal fibula. Conclusions: The current literature provides sufficient arguments for the development of an evidence-based surgical treatment algorithm for unstable syndesmotic injuries.

## 1. Introduction

An increasing interest in the treatment of unstable syndesmotic injuries during the last decade has led to an enormous amount of literature not easy to review [1,2,3,4,5,6,7,8,9,10,11,12,13,14,15,16,17,18,19,20,21,22,23,24,25,26,27,28,29,30,31,32,33,34,35,36,37,38,39,40,41,42,43,44,45,46,47,48,49,50,51,52,53,54,55,56,57,58,59,60,61,62,63,64,65,66,67,68,69,70,71,72,73,74,75,76,77,78,79,80,81,82,83,84,85,86,87,88,89,90,91,92,93,94,95,96,97,98,99,100,101,102,103,104,105,106,107,108,109,110,111,112,113,114,115,116,117,118,119,120,121,122,123,124,125,126,127,128,129,130,131,132,133,134,135,136,137,138,139,140,141,142,143,144,145,146,147,148,149,150,151,152,153,154,155,156,157,158,159,160,161,162,163,164,165,166,167,168,169,170,171]. Syndesmosis as a search term in PubMed, for example, currently revealed 1271 results as of 28 October 2021.

The ligaments stabilizing the inferior tibiofibular syndesmosis prevent excess fibular motion in multiple directions: anterior-posterior translation, lateral translation, cranio-caudal translation, and internal and external rotation [6,62,63,68,79,164]. Appropriate fibular position and limited rotation are necessary for normal syndesmotic function and talar position within the ankle mortise [66]. Reconstruction of unstable syndesmotic injuries is not trivial, and there are no generally accepted treatment guidelines [18,63,80,119]. Thus, there still remain considerable controversies regarding diagnosis, classification, and treatment of unstable syndesmotic injuries [63,119,149]. The most common controversies regarding surgical treatment are related to screw fixation versus dynamic fixation, the use of reduction clamps, open versus closed reduction, and the role of the posterior malleolus and of the anterior inferior tibiofibular ligament (AITFL).

Although several studies have clearly shown serious problems concerning the use of syndesmotic screws, this method is still considered the gold standard for treatment of unstable syndesmotic injuries by the majority of orthopedic and trauma surgeons [10,12,18,19,24,27,30,39,51,53,64,70,148,159]. This controversy may be explained by an obvious lack of further education and training, by the misconception that this surgical technique is simple and easy to perform, and by the fact that syndesmotic screws still represent the cheapest solution, at least considering the short term [115].

Alternative methods for treatment of unstable syndesmotic injuries reported in literature include syndesmotic hooks or hook plates [45,160,166], transfixation bolts [162], various suture button constructs [5,8,38,94,95,106,117,120,131,132,141,142,165], ligament bracing [58,80,98,119,135,140], tendon autograft [25,56,91,98] or even allograft reconstruction [35]. During the last decades, several studies have clearly shown superior results after flexible dynamic syndesmotic stabilization compared to the use of syndesmotic screws with regard to accuracy of reduction, functional outcome, and even development of posttraumatic ankle arthritis [9,32,34,48,57,72,76,94,111,124,134,171]. 

Therefore, our aim was to draw important conclusions from the pertinent literature concerning surgical treatment of unstable syndesmotic injuries, to transform these conclusions into surgical principles supported by literature, and finally to fuse these principles into an evidence-based surgical treatment algorithm. 

## 2. Materials and Methods

From 1 October 2019 to 28 October 2021, the first and senior authors (M.R. and C.E.) independently searched PubMed, Embase, Google Scholar, The Cochrane Database of Systematic Reviews, and the reference lists of systematic reviews of relevant studies dealing with the surgical treatment of unstable syndesmotic injuries. The mentioned databases were selected because they represent the prevalent and generally accepted databases used for medical research. The following search terms listed in alphabetical order were used: anatomic reduction, anatomic repair, ankle fracture, anterior inferior tibiofibular ligament (AITFL), augmentation, diastasis screw, flexible stabilization, high ankle sprain, InternalBrace, ligament bracing, positioning screw, posterior malleolus, rotational stability, suture button, syndesmo*, syndesmosis, syndesmotic screw, tibiofibular*, tightrope, and treatment algorithm. Only articles published in English, German, or Spanish language were included. Other than language, there were no further restrictions for the inclusion of articles. Letters to the editors, short comments, incomplete, or inaccessible full-text articles were excluded. Included articles were assigned to the following main topics:Importance of anatomic reductionClosed versus open reduction under direct visualizationRole of reduction clampsRole of syndesmotic screwsRole of flexible dynamic stabilization techniquesRole of the AITFLRole of the posterolateral malleolus

After assessment of the scientific quality of the assigned articles, we tried to draw important conclusions from the pertinent literature concerning the surgical treatment of unstable syndesmotic injuries. Then, we transformed these conclusions into clinically relevant surgical principles supported by literature, and finally, we fused these principles into a surgical treatment algorithm.

## 3. Results

A total of 171 articles were included for further considerations [1,2,3,4,5,6,7,8,9,10,11,12,13,14,15,16,17,18,19,20,21,22,23,24,25,26,27,28,29,30,31,32,33,34,35,36,37,38,39,40,41,42,43,44,45,46,47,48,49,50,51,52,53,54,55,56,57,58,59,60,61,62,63,64,65,66,67,68,69,70,71,72,73,74,75,76,77,78,79,80,81,82,83,84,85,86,87,88,89,90,91,92,93,94,95,96,97,98,99,100,101,102,103,104,105,106,107,108,109,110,111,112,113,114,115,116,117,118,119,120,121,122,123,124,125,126,127,128,129,130,131,132,133,134,135,136,137,138,139,140,141,142,143,144,145,146,147,148,149,150,151,152,153,154,155,156,157,158,159,160,161,162,163,164,165,166,167,168,169,170,171]. Among them, 47 articles concerned syndesmotic screw fixation and 41 dynamic fixations of the syndesmosis. Twenty-five studies compared screw fixation with dynamic fixations, and seven out of these comparisons were randomized controlled trials [9,32,34,72,76,111,124]. Nineteen articles addressed the posterior malleolus, 14 the role of the AITFL, and eight the use of reduction clamps.

### 3.1. Importance of Anatomic Reduction

Malreduction of the distal tibiofibular syndesmosis still remains a common complication associated with the surgical management of ankle fractures [23,31]. Syndesmotic malreduction can lead to severe alterations in the biomechanics of the ankle and thereby to chronic pain and premature degenerative changes of the ankle joint [31]. Therefore, ankle fractures with need for syndesmotic stabilization are still associated with a high rate of secondary osteoarthritis [118]. As early as 1961, Willenegger stated that posttraumatic osteoarthritis of the ankle almost always is due to an incongruity between the ankle mortise and the talus and not due to an intraarticular fracture itself [162]. He reported on 31 of 32 cases that rapidly developed severe posttraumatic ankle arthritis even after only slight malreduction of the ankle mortise [162]. Several well-known studies confirmed Willenegger’s early observations, and today, it is generally accepted that even a small syndesmotic displacement of less than 1 mm can have devastating consequences for the rapid development of posttraumatic ankle arthritis [1,59,62,65,100,102,111,116,123,153]. Significant increases in tibiotalar contact pressures occur when external rotation stresses are added to axial loading in an unstable tibiofibular syndesmosis. Moderate and severe syndesmotic injuries are associated with a significant increase in mean contact pressure combined with a shift in the center of pressure and rotation of the fibula and talus. In this context, simple syndesmotic injuries represent partial ruptures without signs of instability in the clinical and radiological evaluation compared to moderate and severe syndesmotic injuries. Moderate syndesmotic injuries show clinical signs of dynamic instability without static dislocation in the radiographic evaluation, and severe injuries show signs of dislocation of the fibula in the incisura tibiofibularis even in the static examinations.

Considerable changes in ankle joint kinematics and contact mechanics may explain why unstable syndesmotic injuries take longer to heal and are more likely to develop long-term dysfunction and ankle arthritis [65]. Moreover, in the long run, post-traumatic ankle osteoarthritis in known to have a large and negative impact on the patients´ quality of life [172]. The single most important prognostic factor after unstable injury of the distal tibiofibular syndesmosis with or without fracture is anatomic reduction of the distal fibula and fitting into the tibial incisura [112]. Therefore, anatomic reduction of an injured syndesmosis is crucial for an optimal long-term clinical result [7,13,31,151]. In this context, anatomic reduction is defined as complete restoration of the physiological anatomy in the first place but does also imply recovery of the physiologic tension of the repaired ligamentous structures.

Main conclusion: Anatomic reduction is crucial for the long-term results regarding functional outcome and development of posttraumatic ankle osteoarthritis.

### 3.2. Closed vs. Open Reduction under Direct Visualization

To reliably achieve anatomic reduction of the syndesmosis, open reduction of the AITFL under direct visualization has been advocated, as there is still a lack of appropriate examination techniques to confirm anatomic closed reduction during surgery [26,87,98,103,119,127,143,150]. Tornetta showed that the anterolateral articular surface of the distal tibia to the anteromedial fibular articular surface is an accurate visual landmark for anatomic reduction of the syndesmosis [143]. Another advantage of open reduction under direct visualization is the opportunity to avoid posttraumatic anterolateral impingement syndrome by removing torn parts of the AITFL out of the joint. And aside from that, we should remember that the main goal in the treatment of unstable syndesmotic injuries is to avoid osteoarthritis and not scars. 

Main conclusion: Open reduction by direct visualization is strongly recommended.

### 3.3. Role of Reduction Clamps

Several studies and reports have clearly shown that it is possible and even highly likely to over-compress the syndesmosis when using reduction clamps or forceps [26,36,60,82,85,107,121,156]. In this context, over-compression is defined as any sign of displacement of the talus out of the mortise due to compression of the fibula against the tibia even in correct position of the fibula within the incisura. Haynes et al. demonstrated a significant correlation between increased clamp forces and syndesmotic over-compression and determined objective forces that lead to over-compression of the syndesmosis [60]. Another study by Miller demonstrated that intraoperative clamping and fixation can cause statistically significant malreduction of the syndesmosis [85]. This should alert clinicians that clamp and screw placement can cause iatrogenic malreduction of the syndesmosis. These dangers occur with specific clamp and screw angles in particular. Mahapatra reported that over-compression of the syndesmosis can even cause significant subluxation of the talus [82]. Therefore, care should be taken to avoid over-compression by use of reduction clamps, as this may affect ankle motion and functional outcomes. Cadaver experiments by Phisitkul et al. showed that clamp placement in the neutral anatomical axis reduced the syndesmosis most accurately, but nevertheless, over-compression was frequently observed. Placing the clamp obliquely malreduced the unstable syndesmosis [107]. Furthermore, based on the results of his recent cadaveric study, Rushing stated that inherent variabilities in the applied clamp force by surgeons appear to contribute to the unacceptably high coronal syndesmotic malreduction rate [121]. Goetz et al. have shown that Achilles tension mitigates fibular malalignment measured in cadaveric studies of syndesmotic clamping [53].

In our experience, it seems to depend on the stability of the posterior malleolus and the medial and lateral collateral ankle ligaments if significant over-compression with consecutive dislocation of the talus is possible or not, but this theory has not been proven so far. Cherney described that a stable posterior malleolus does not have a protective effect against over-compression [26]. In our opinion, this might be due to unrecognized additional injuries to the collateral ankle ligaments. Additionally, from an anatomic point of view, there is no physiologic dynamic force leading to displacement of the syndesmosis and therefore needing neutralization by a reduction clamp. As early as 1953, Costigan stated that reduction of diastasis of the ankle mortise is a very simple procedure, not requiring a reducer or any other form of mechanical device [37]. Therefore, at least in acute cases, the surgeon should prefer to analyze the real reasons preventing easy reduction of the syndesmosis instead of increasing the force of reduction clamps.

Main conclusions: There is a high risk for over-compression and malreduction of the syndesmosis by use of reduction clamps or forceps. Therefore, at least in acute cases, the use of reduction clamps or forceps should be avoided whenever possible.

### 3.4. Role of Syndesmotic Screws

Syndesmotic screws, also referred to as diastasis screws, situational screws, transfixation screws, trans-syndesmotic screws, or positioning screws, have been used at least since 1947 [37]. Therefore, surgical treatment with syndesmotic screw fixation has been performed for several decades now, and this method is still considered the gold standard of treatment of unstable syndesmotic injuries by the majority of orthopedic and trauma surgeons [5,61,67,125,151,159,162,163,169,170]. However, there is still an ongoing discussion concerning ideal reference points and anatomic landmarks for optimal positioning of syndesmotic screws [74,75]. The risk of malpositioning of syndesmotic screws is very high, and a lack of standard radiological or physical references for accurate syndesmotic screw placement is a potential contributing factor in syndesmotic screw malpositioning [75]. Furthermore, several studies have clearly shown serious problems concerning the use of syndesmotic screws due to malreduction of the ankle mortise or high complication rates after syndesmotic screw removal, for example, [10,12,24,39,51,54,70,100,123,148]. Ovaska analyzed patients with malreduced ankle fractures undergoing re-operation and showed that the most common indication for re-operation was syndesmotic malreduction in 59% of cases [100]. In another series of 160 consecutive patients who underwent syndesmosis screw fixation, 13 patients needed revision surgery. Among them, the incidence of recurrent diastasis of the ankle mortise was 92% [12]. Gardner evaluated twenty-five patients with ankle fractures and syndesmotic instability who had open reduction and syndesmotic fixation [51]. A total of 52% of syndesmoses were malreduced on CT scan but went undetected by plain radiographs. Radiographic measurements did not accurately reflect the status of the distal tibiofibular joint in this series of ankle fractures. Furthermore, post-reduction radiographic measurements were inaccurate for assessing the quality of the reduction [51]. In a prospective randomized controlled multicenter trial with 103 patients, the rate of malreduction using screw fixation was 39% [124]. Therefore, especially due to the repeatedly reported high risk of malreduction, the first-line use of syndesmotic screws as a gold standard for the treatment of unstable syndesmotic injuries should be reconsidered. 

Main conclusions: The first-line use of syndesmotic screws as a gold standard for the treatment of unstable syndesmotic injuries should be reconsidered. Syndesmotic screws should be used only as a salvage procedure.

### 3.5. Role of Flexible Dynamic Stabilization Techniques

The currently emerging surgical techniques for flexible dynamic stabilization of the syndesmosis are not new regarding their principles, but the available implants provided by industry enabling low-risk and straightforward surgery have significantly improved over the past two decades [5,8,9,33,95,139].

As early as 1961, Willenegger already stated that the preservation of the natural elasticity of the syndesmosis is of greatest importance not only for the final results but also for the initial postsurgical treatment. Accordingly, he refused the use of the rigid transfixation bolt [162]. However, at that time, there was a lack of suitable alternative implants. In 1955, Schumann developed a kind of early precursor of suture-button constructs consisting of two steel plates placed over the medial and lateral malleolus connected via a transmalleolar tensioning wire bolt [131]. This indicates that even at that early time, surgeons were searching for flexible alternatives to the rigid screw or bolt fixations. Some decades later, in 1991, Seitz et al. reported on the repair of the tibiofibular syndesmosis with a flexible implant consisting of a double thickness of No. 5 braided polyester suture tied over polyethylene buttons situated medially and laterally [132].

The currently most widely-used flexible dynamic stabilization device is the knotless syndesmosis TightRope^®^ implant system produced by Arthrex (Naples, FL, USA). Recently, this device has attracted a great deal of even public attention due to an extremely accelerated return to play in high-level athletes after surgical stabilization of high ankle sprains [104]. Tightrope surgeries have been brought into the spotlight known as “Tua surgery” by the circumstances of the Alabama Crimson Tide’s 2018 football season. Starting quarterback Tua Tagovailoa suffered a high ankle sprain on 1 December 2018, and the following day, his injured right ankle was stabilized by Norman Waldrop using two knotless syndesmosis TightRope^®^ implant systems. Tagovailoa returned to play just 27 days after surgery to lead the Alabama Crimson Tide football team to a win over Oklahoma in the College Football Playoff semifinal on 29 December 2018.

During the last decades, several studies have clearly shown superior results after flexible dynamic syndesmotic stabilization compared to the use of syndesmotic screws with regard to accuracy of reduction, functional outcome, and even development of posttraumatic ankle arthritis [9,29,32,34,48,49,55,57,69,71,72,76,83,94,102,111,115,124,125,129,130,132,138,171]. Seven of these comparisons were randomized controlled trials [9,32,34,72,76,111,124]. In 2019, for example, Sanders showed that the rate of malreduction using screw fixation was 39% compared with 15% using TightRope^®^ fixation, and the reoperation rate was much higher in the screw group compared with TightRope^®^ (30% vs. 4%, *p* = 0.02) with the difference driven by the rate of implant removal [124]. With deliberate malreduction in a cadaver model, Westermann reported that suture-button fixation of the syndesmosis results in less post-fixation displacement compared with screw fixation. The suture button’s ability to allow for natural correction of deliberate malreduction was greatest with posterior off-axis clamping [161]. Therefore, dynamic syndesmotic fixation may even mitigate clamp-induced malreduction. A systematic review of suture-button versus syndesmotic screw in the treatment of distal tibiofibular syndesmosis injury by Zhang et al. showed that the suture-button device could lead to better objective range of motion measurements and earlier return to work. Besides, the suture-button fixation group had lower rates of implant removal, implant failure, and malreduction [171]. 

A randomized trial by Anderson et al. comparing suture-button with single syndesmotic screw for syndesmosis injury attested that patients treated with a suture-button device had higher AOFAS scores, OMA scores, and EQ-5D Index scores as well as better VAS scores for pain during walking and pain during rest. Moreover, the suture-button group had less widening seen radiographically at two years than did the patients in the syndesmotic screw group [9]. Five years after syndesmotic injury treated with either suture-button or syndesmotic screw within a randomized controlled trial, Ræder et al. found better AOFAS and OMA scores and even lower incidence of ankle osteoarthritis in the suture-button group [111]. 

These long-term results clearly favor the use of suture-button devices when treating an acute syndesmotic injury. In a recent meta-analysis performed by Shimozono, the suture-button technique resulted in improved functional outcomes as well as lower rates of broken implant and joint malreduction. Based on these findings, the suture-button technique warrants a grade A recommendation by comparison with the syndesmotic screw technique for the treatment of syndesmosis injuries [134]. 

Main conclusion: Use flexible dynamic stabilization techniques for treatment of unstable syndesmotic injuries whenever possible.

### 3.6. Role of the AITFL

According to the aforementioned literature, it is possible to achieve very good results in the treatment of unstable syndesmotic injuries using suture-button devices in most cases. However, a study by Clanton recently showed that in some cases, it might not be possible to sufficiently stabilize sagittal translation of the distal fibula by one TightRope^®^ and especially rotary instability even by use of two such flexible dynamic suture-button devices [29]. These observations were confirmed in a biomechanical study by Goetz et al., who reported that flexible trans-syndesmotic fixation alone was found to be insufficient for restoring rotational stability to the ankle or preventing sagittal plane displacement of the distal fibula, and thus, repairs to simulate anatomic structures disrupted during a syndesmosis injury were required to restore rotational stability [55]. Moreover, Clanton showed that isolated injuries to the AITFL resulted in the most substantial reduction of resistance to external rotation and that even isolated injuries to the AITFL alone may lead to significant external rotary instability of the ankle mortise [29]. Recently, these findings were confirmed by a cadaveric robotic study by Patel, who showed that ankle instability is similar after both isolated AITFL and complete syndesmosis injury and can persist after suture-button fixation in the sagittal plane in response to an inversion stress [105]. 

We have also experienced these problems clinically, and therefore, we started to augment our repairs of the AITFL by use of an *Internal*Brace^TM^ (Arthrex, Naples, FL, USA) in 2013 [119]. Shoji, Teramoto, and Hajewski published a similar surgical technique for ligament augmentation of the AITFL in 2018 and 2019, respectively [58,135,140]. 

Even earlier, Nelson reported in 2006 on the importance of the AITFL and described two methods for an anatomic repair or reconstruction of the AITFL [98]. However, at that time, there were no suitable implants available, so that this technique was not widely adopted. This article by Nelson was a report of 50 unselected consecutive unstable ankle fractures in which he specifically visually examined the injured AITFL and described a technique of direct visual reduction and flexible repair of the syndesmotic disruption using autograft or sutures secured to bone with traditional screws. Syndesmotic screw fixation of transmalleolar ankle fractures was not necessary in any of the 50 cases when the AITFL was repaired directly by the techniques described. In this unselected consecutive series of patients, Nelson documented a 100% incidence of injury of the AITFL independent of the fracture classification (Weber A, B, and C fractures were included in the study) [98]. The total incidence of bony avulsions of the AITFL was 26%, consisting of an 18% incidence of fibular avulsions (Wagstaffe fragments) and an 8% incidence of tibial avulsions (Tubercule de Chaput fragments). Based upon that clinical experience and continuing interest in the syndesmotic ligaments, it was Nelson’s conviction that the AITFL is ruptured in virtually 100% of unstable ankle fractures and that direct visual reduction and repair of ligament disruptions and bone avulsions results in reliable restoration of ankle anatomy and stability. Consequently, Littlechild advocated that consideration should also be given to reconstruction of the AITFL to augment the syndesmosis fixation, which may provide a stronger restoration of ankle stability compared to repairing the posterior inferior tibiofibular ligament (PITFL) in isolation, for example, by fixation of a posterior malleolus avulsion fracture [77]. In total, we found 13 publications recommending open repair and augmentation of the AITFL [2,11,20,28,58,77,80,98,119,135,140,167,170]. 

Main conclusions: The AITFL is an important stabilizer, especially for rotary stability, and even isolated injuries to the AITFL alone may lead to significant external rotary instability of the ankle mortise. Therefore, an unstable AITFL should be repaired and augmented. Bony avulsion fragments can obstruct anatomic reduction of the distal tibiofibular joint but when reduced anatomically can serve as landmarks for reduction.

### 3.7. Role of the Posterior Malleolus

Sir Astley Cooper, in 1822, first described fractures of the ankle with involvement of the posterior lip of the tibia [97]. The posterior lip of the distal tibia was entitled the “posterior malleolus” by Destot in 1911, and Henderson and Stuck, in 1935, suggested the term “trimalleolar” for ankle fractures involving the medial and lateral malleolus and the posterior lip of the tibia [97]. In 1922, Lounsberry and Metz discussed trimalleolar fractures as well as those with involvement of the anterior tibial lip, and they were the first to advocate open reduction and internal fixation of displaced posterior and anterior tibial fragments [97].

The posterior malleolus is affected in around 40% of ankle fractures [146]. Anatomical reduction of the articular surface and fibular notch are essential for ankle stability and functional outcomes. To address the posterior malleolus when treating ankle fractures, surgeons should choose the most adequate approach based on the fracture pattern and their own experience [3,14,15,16,17,99,146,152,154,155]. Anatomical reduction and stable fixation of the posterior malleolus are critical to improve outcomes [1,3,14,15,16,17,21,43,46,50,84,86,99,144,146,153,154]. 

It was Gardner who first observed, in 2006, that syndesmotic stability may be obtained more effectively by fixation of the posterior malleolus rather than by using a syndesmotic screw [50]. This observation was later confirmed by several authors [17,84,86,144].

Tosun et al. compared posterior malleolus versus syndesmotic screw fixation in trimalleolar ankle fractures. The results of this study demonstrate that posterior malleolar fracture fixation is closely related to successful radiological and functional outcomes after trimalleolar fractures. Syndesmotic screw fixation may not be needed in cases in which the posterior malleolar fracture has been fixated. For these reasons, the authors recommended that all posterior malleolar fractures have to be fixed regardless of size [144].

Furthermore, according to Verhage, the posterior fragment size is not a clear indication for its fixation [153]. A step-off, however, seems to be an important indicator for developing posttraumatic osteoarthritis and worse functional outcome [1,21,43,153]. Therefore, displaced posterior fragments involving the intra-articular surface need to be reduced and fixated to prevent postoperative persisting step-off [153]. Furthermore, direct fixation of the posterior malleolus via an open posterolateral approach seems superior to percutaneous anterior-to-posterior fixation [15,16,17,84,99,154,155]. 

When posterior malleolus fractures occur with syndesmotic injury, anatomic fracture reduction and fixation are paramount, as they can affect syndesmotic reduction, especially with larger fragments [46]. Therefore, in any case of a displaced fracture of the posterior malleolus with a bone fragment big enough for screw or plate fixation and with an intact PITFL, the posterior malleolus should be considered the key for anatomic reduction of the syndesmosis [46]. After fixation of the posterior malleolus in an anatomic position, the further surgical steps for complete reduction and stabilization of the syndesmosis will be easy to perform. Therefore, open reduction and direct fixation of the posterior malleolus should be performed as the first step [84,97]. This surgical order has the additional advantage that fluoroscopic control of reduction quality is not limited by other implants, such as a fibular plate. In case of fixation of the posterolateral malleolus in malposition, anatomic reduction of the syndesmosis will not be possible anymore due to ligamentotaxis. Therefore, in cases where it is very difficult or even impossible to fix the posterior malleolus in an anatomic position, it might be better not to fix the posterior malleolus than to fix it in malposition.

Main conclusions: A step-off of the posterior malleolus is an important indicator for developing posttraumatic osteoarthritis and worse functional outcome. Therefore, displaced posterior malleolar fractures have to be fixed regardless of size. Fix the posterior malleolus directly from posterior whenever possible. When fixing the posterior malleolus, start with this procedure, as the posterior malleolus is the key for anatomic reduction of the syndesmosis.

As a kind of summary of our investigations, we formulated the following principles for the surgical treatment of unstable syndesmotic injuries, which can be considered supported by varying degrees of evidence in literature:

### 3.8. Recommended Principles for the Surgical Treatment of Unstable Syndesmotic Injuries

Anatomic reduction is crucial for the long-term results;Open reduction by direct visualization is strongly recommended;Repair what is injured;Bony avulsion fragments must be identified and reduced (when big enough) and can serve as landmarks for anatomic reduction;The use of reduction clamps or forceps should be avoided whenever possible;Fix the posterolateral malleolus directly from posterior whenever possible;When fixing the posterolateral malleolus, start with this procedure;The AITFL is an important stabilizer especially for rotational stability;An unstable AITFL should be repaired and augmented;Use flexible dynamic stabilization techniques whenever possible; andUse syndesmotic screws only as a salvage procedure.

These recommended main principles for the surgical treatment of unstable syndesmotic injuries were fused into the following evidence-based surgical treatment algorithm (Figure 1). Practicability of this algorithm is additionally based on the clinical experience of the first author (M.R.), who has performed more than 300 flexible dynamic syndesmotic stabilizations using an *Internal*Brace^TM^ since 16 December 2013. Corresponding clinical and radiological outcome studies are currently running.

### 3.9. Evidence-Based Surgical Treatment Algorithm for Unstable Syndesmotic Injuries

For the first decision before starting surgery, the surgeon has to evaluate if there is a displaced posterior malleolar fragment amenable to open anatomic reduction and direct fixation with a lag screw or an anti-glide plate (Figure 1A). If so, this procedure should be performed as the first step of surgery due to the following three reasons: (1) direct fixation provides excellent primary stability of the posterior syndesmosis. (2) After anatomic reduction and direct fixation of the posterolateral malleolus, the following surgical steps will be quite easy to perform. (3) Fluoroscopic control of reduction quality is not limited by other implants like a fibular plate.

The next step after direct fixation of the posterior malleolus, or the first step in case of an intact posterior malleolus, is direct visualization of the AITFL and an intraoperative grading of the injury pattern under direct view (Figure 2). The surgeon should be aware that only grade 3 injuries would be detected by the well-known Cotton test or its hook test modifications [22,90,101,103]. Increased external rotation and posterior sagittal translation indicates an isolated tear or bony avulsion of the AITFL, representing a grade 1 injury (Figure 2A); increased anterior sagittal translation indicates an additional tear or bony avulsion of the PITFL, representing a grade 2 injury (Figure 2B); and increased lateral translation indicates additional medial instability, representing a grade 3 injury (Figure 2C). Further treatment strategy should be according to the detected individual injury pattern:Grade 1: Single anterior stabilization with an *Internal*Brace^TM^ (Figure 1B);Grade 2: Double stabilization = anterior and posterior stabilization (Figure 1C); andGrade 3: Triple stabilization = double stabilization + central tightrope or syndesmotic screw (Figure 1D).

We have already exemplified in detail how to perform these stabilization techniques in 2017 [119]. In our experience, the most common procedure is anterior stabilization with an *Internal*Brace^TM^, either as single anterior stabilization or as double stabilization in combination with direct refixation of the posterior malleolus or a slightly posteriorly directed suture-button device in order to indirectly stabilize an injured PITFL. As a promising alternative to an indirect posterior stabilization by use of a suture-button device, recently, some surgeons have also been performing a direct stabilization of the PITFL with an additional *Internal*Brace^TM^ to an increasing degree. However, to the best of our knowledge, this technique has only been described in a cadaver model so far [119]. Triple stabilization by use of an additional central suture-button device or a syndesmotic screw proximal of the incisura is rarely necessary in our hands. As a suture-button-device, we have used the knotless syndesmotic TightRope^®^ implant system provided by Arthrex in our own cases, as this implant, to the best of our knowledge, represents the most frequently used and most investigated suture-button-device. However, alternative implant systems provided by other manufacturers are currently available as well.

A clinical example of a single anterior stabilization in a type B ankle fracture with displaced bony tibial avulsion of the AITFL (Figure 3A) is shown in Figure 3. After secure radiological and clinical exclusion of any unstable injury to the PITFL or the posterior malleolus, the distal fibular fracture was fixed with a special anatomically shaped titanium distal fibular plate with eyelets for tape augmentation of the AITFL and PITFL (Arthrex, Naples, FL, USA). The bony AITFL avulsion fragment was anatomically reduced under direct visualization and fixed with a 3.5-mm headless compression screw (Figure 3B). Due to its size, the bony fragment was not amenable to fixation with a stronger screw. Therefore, a FiberTape^®^ (Arthrex, Naples, FL, USA) for augmentation of the AITFL was pulled through the anterior eyelets of the plate (Figure 3C) and fixed to the distal tibia exactly in line with the course of the uninjured AITFL with a 4.75-mm SwiveLock^®^ (Arthrex, Naples, FL, USA). Figure 3 D shows the final result after osteosynthesis of the distal fibula with a titanium plate, refixation of the bony avulsion fragment with a compression screw, and augmentation of the AITFL with an *Internal*Brace^TM^ (Arthrex, Naples, FL, USA).

## 4. Discussion

Our aim was to provide the currently best available evidence for surgical treatment of unstable syndesmotic injuries. Therefore, we searched PubMed, Embase, Google Scholar, The Cochrane Database of Systematic Reviews, and the reference lists of systematic reviews of relevant studies dealing with the surgical treatment of unstable syndesmotic injuries. Then we tried to draw important conclusions from the pertinent literature concerning surgical treatment of unstable syndesmotic injuries, to transform these conclusions into surgical principles supported by the literature, and finally to fuse these principles into an evidence-based surgical treatment algorithm. In our hands, this surgical treatment algorithm has been working well for more than seven years now.

A total of 171 articles were analyzed, and to summarize our results, we found the following main principles for surgical treatment of unstable syndesmotic injuries: (1) anatomic reduction is crucial for the long-term results; (2) open reduction by direct visualization is strongly recommended; (3) repair what is injured; (4) bony avulsion fragments must be identified and reduced and can serve as landmarks for anatomic reduction; (5) the use of reduction clamps or forceps should be avoided whenever possible; (6) fix the posterolateral malleolus directly from posterior whenever possible; (7) when fixing the posterolateral malleolus, start with this procedure; (8) the AITFL is an important stabilizer especially for rotational stability; (9) an unstable AITFL should be repaired and augmented; (10) use flexible dynamic stabilization techniques whenever possible; and (11) use syndesmotic screws only as a salvage procedure.

However, there are some limitations to our study. Our study was not a systematic review designed according to the Preferred Reporting Items for Systematic Reviews and Meta-Analyses (PRISMA) guidelines, and study quality of the cited references was not assessed based on any quality appraisal scales. Moreover, due to our language selection criteria, there might be additional important studies not considered in our work.

Furthermore, our aim was to develop the best available solution for unstable syndesmotic injuries with regard to medical and not economic issues, and we are aware of the fact that our approach for treating unstable syndesmotic injuries is quite expensive compared to the simple and cheap syndesmotic screws, for example [115]. However, regarding the well-known severe consequences of failed syndesmotic repairs, we strongly recommend treating unstable syndesmotic injuries using the best implants available. In the long run, it does not seem reasonable to save money in the primary care for unstable syndesmotic injuries because the consequence might be very high costs for the treatment of ankle osteoarthritis.

## 5. Conclusions

Current literature provides sufficient arguments for the development of an evidence-based surgical treatment algorithm for unstable syndesmotic injuries. Anatomic reduction is crucial to prevent posttraumatic osteoarthritis. Therefore, flexible dynamic stabilization techniques should be preferred whenever possible. An unstable AITFL should be repaired and augmented, as it represents an important stabilizer of external rotation of the distal fibula.

## Figures and Tables

**Figure 1 jcm-11-00331-f001:**
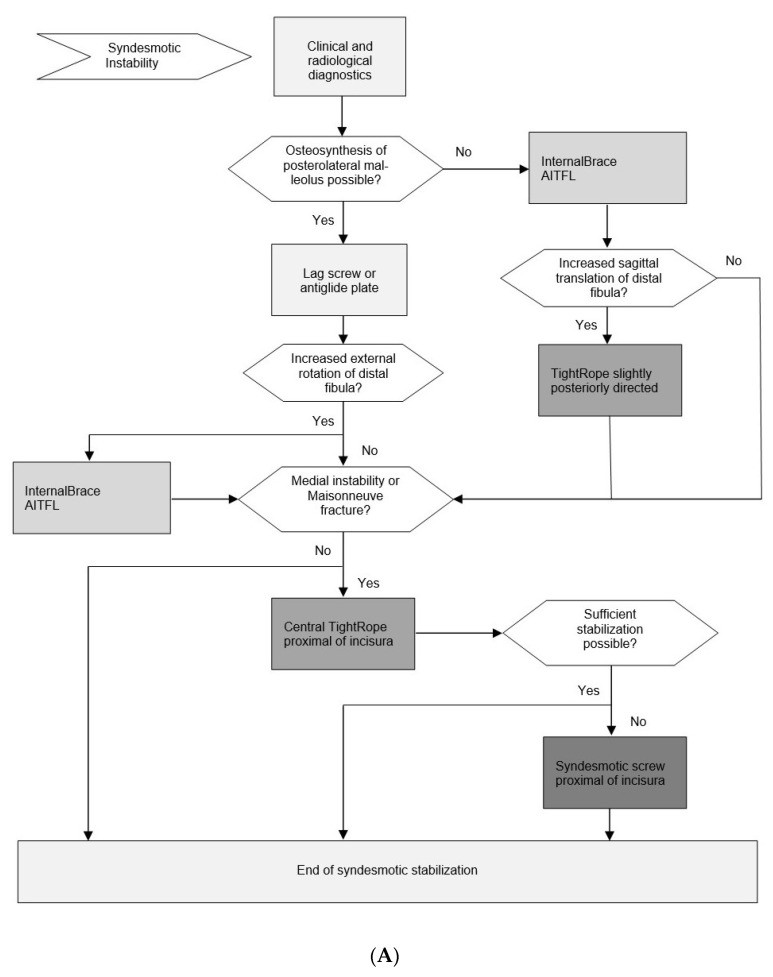
(**A**) Evidence-based surgical treatment algorithm for unstable syndesmotic injuries. (**B**) Evidence-based surgical treatment algorithm for unstable syndesmotic injuries. Exemplary treatment path for grade 1 injuries. (**C**) Evidence-based surgical treatment algorithm for unstable syndesmotic injuries. Exemplary treatment path for grade 2 injuries. (**D**) Evidence-based surgical treatment algorithm for unstable syndesmotic injuries. Exemplary treatment path for grade 3 injuries.

**Figure 2 jcm-11-00331-f002:**
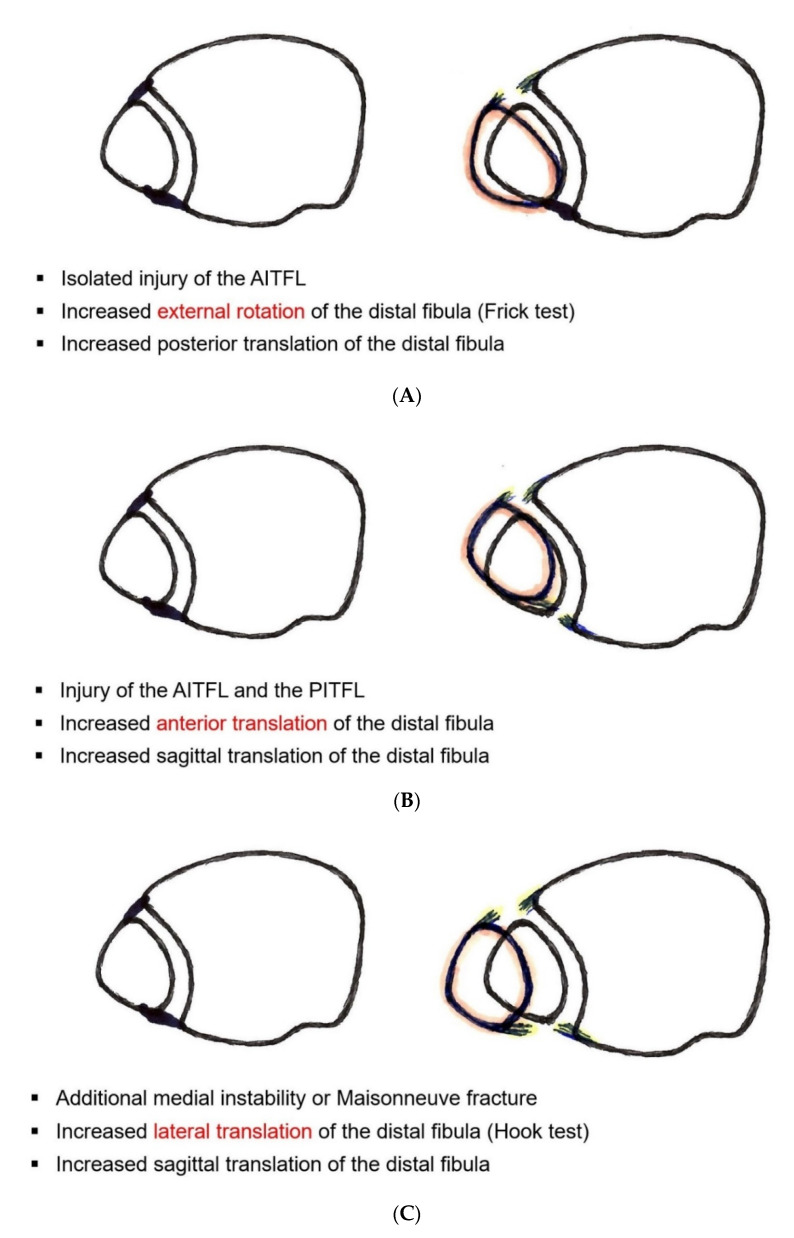
(**A**) Intraoperative assessment of a grade 1 injury. (**B**) Intraoperative assessment of a grade 2 injury. (**C**) Intraoperative assessment of a grade 3 injury.

**Figure 3 jcm-11-00331-f003:**
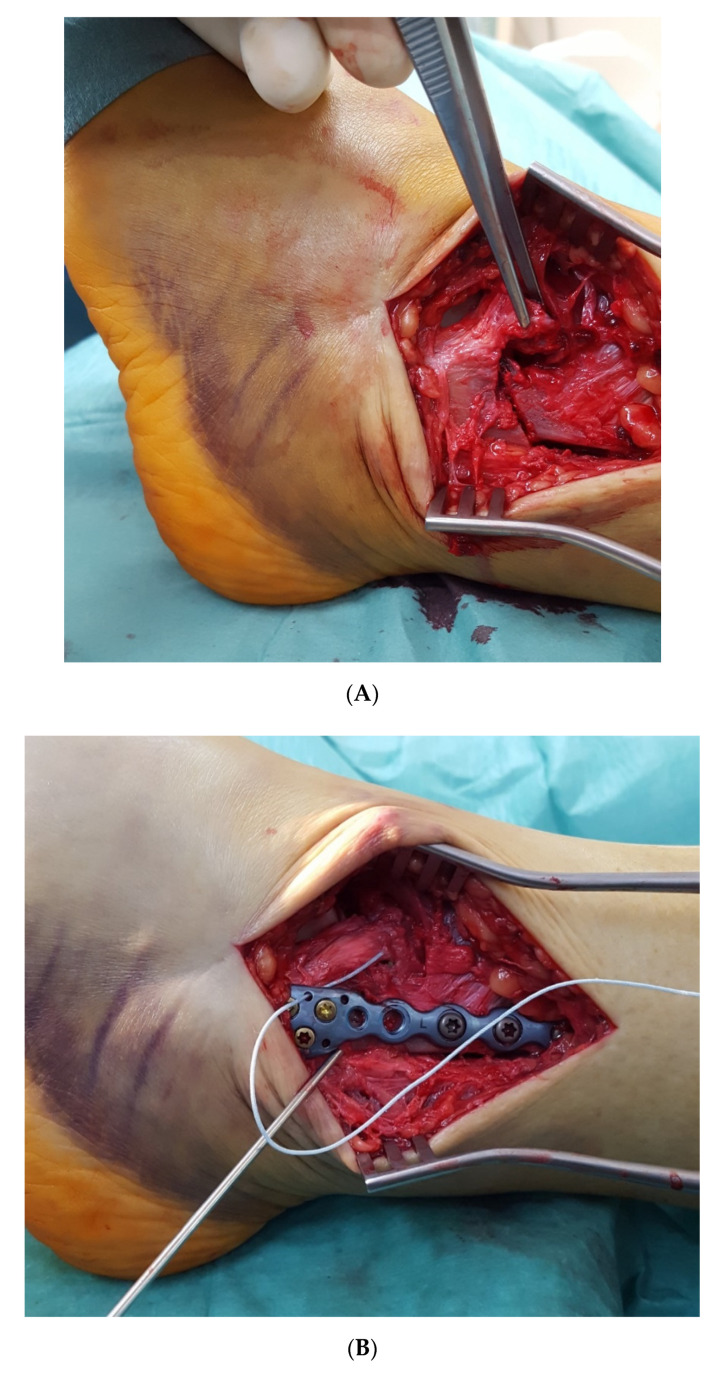
(**A**) Type B ankle fracture with displaced tibial bony avulsion of the AITFL. (**B**) Type B ankle fracture with displaced bony avulsion of the AITFL. The distal fibular fracture was fixed with a special anatomically shaped titanium distal fibular plate with eyelets for tape augmentation of the AITFL and PITFL (Arthrex, Naples, FL, USA). The bony AITFL avulsion fragment was anatomically reduced and fixed with a 3.5-mm headless compression screw. (**C**) Type B ankle fracture with displaced bony avulsion of the AITFL. A FiberTape^®^ (Arthrex, Naples, FL, USA) for augmentation of the AITFL has been pulled through the anterior eyelets of the plate and fixed to the distal tibia with a 4.75-mm SwiveLock^®^ (Arthrex, Naples, FL, USA). (**D**) Type B ankle fracture with displaced bony avulsion of the AITFL. Final result after osteosynthesis of the distal fibula with a titanium plate, refixation of the bony avulsion fragment with a compression screw, and augmentation of the AITFL with an *Internal*Brace^TM^ (Arthrex, Naples, FL, USA).

## Data Availability

Not applicable.

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
