# Peer review of "Evidence-Based Surgical Treatment Algorithm for Unstable Syndesmotic Injuries"

_jcm, 2022, doi:10.3390/jcm11020331_

Round 1
Reviewer 1 Report
An interesting work, a subject that has not been fully explored and fully explained so far, i.e. the surgical treatment algorithm for unstable syndesmotic injuries. The introduction is well written, presenting the outline of the literature and the goals of the work. Materials and Methods: Please expand this paragraph further. Please explain why these databases were selected for searching for articles. Please enter the number of analyzed articles for the following main topics. In the discussion, please add a paragraph that briefly summarizes the results. Several key algorithms for the procedure.Author Response
Please see the attachment.

Reviewer 2 Report
Thanks for this thorough literature review. I have several concerns with it. In the abstract you mention that a systematic review for the surgical treatment was performed. However, no clear exclusion criteria were set. In addition, isolated syndesmotic injuries are compared with trimalleolar fractures.
To investigate the surgical treatment of syndesmotic injuries is important. However, the indication for surgery is essential i.e. isolate syndesmotic injury, partial syndesmotic injury or in combination with ankle fracture. This review is too broad and should start with a clear focus on one type of injury.
Please find my concerns below:
Line 50: remainS
Lien 60: Please include a reference. As long as you leave the screw in, I totally agree, but when you consider to take it out after couple of weeks then the re-surgery would exceed the implant costs, although this is covered by the insurance and can be performed under local anesthesia.
Line 85: This seems to be too broad if you consider all articles to be included. At least letter to the editors, short comments, incomplete or unaccessible full text article should have been excluded.
So I assume it was not a systematic review based on your methods. Otherwise please state on this? PRISMA guidelines? Clear exclusion criteria.
Line 120: How do you define moderate and severe syndesmotic injuries?
Line 130: What is anatomic reduction? I.e. is an overstuffed syndesmosis still anatomically reduced?
Line 158: Based on the literature what is the definition of overcompression.
I do agree that reduction clamps or forceps may lead to malreduction or overcompression. However, it has to been differentiated between drilling the hole for the screw or endobutton suture device or placement. Personally I believe it is highly recommended when drilling although this may cause malreduction.
Line 133. Typically the anterior tibiofibular ligament is being reduced under direct visualization. It is difficult to see the other two parts of the syndesmosis.
Furthermore, in my opinion it is essential where to place the sydnesmostic screws: 1 cm above the syndesmosis, or just above it. One or two screws.
Line 208: I would not agree on that sydesmostic screws should no longer be considered as gold standard. The risk of overstuffing/ overcompression of endobutton suture devices is much higher.
Line 245: Does the reoperation rate include the screw removal?
Line 255: How was implant failure defined. How can you recognize this on radiography.
Line 260: This could also be caused by overcompression or too early removal of the screw.
Line 271: What about overcompression?
Line 281: including biomechanical studies is interesting however in the recent literature it is obvious that although in biomechanical laboratories implants are superior this may not be required in vivo.
Line 290: Why would an internal brace give more stability than an endobutton suture device.
Line 301: posterior malleolus? Medial one? Please specify.
Line 310: Do you mean restoration of the avulsion fractures or the ligaments?
Line 315: Did the studies differentiate between acute and chronic syndesmostic injuries?
Line 317: the AITFL is an important stabilizer. I think you already outlined this earlier.
Line 323-325: Certainly this is interesting however has nothing to do with syndesmotic injuries.
Line 361: Reduction of the posterior malleolus should be first step? Do you mean in ankle fixation?
Line 372 373 are contradictory.
Line 382: Bony avulsions can be repositioned and fixed when big enough.
Line 384: For drilling as well?
Line 388: Not only the AITFL but also the PTFL and the IOL are important for stability.
Line 390: What is a salvage procedure?
Line 393: Good to know. But have you published any data yet?
Why do you recommend lag screws or antiglide plates?
Figure 1A is not easy to understand. If I use an internal brace and there is still some medial instability I am supposed to insert an additional tightrope and if both do not provide enough stabilization I am supposed to out in a syndesmostic screw.
Do you recommend Internal brace in all isolated AITFL ruptures without injuries of the IOL and PITFL?
Figure 1: My concerns are if you insert an internal brace first to reduce the AITFL and overstuff it, you will get a malrotation of the remaining syndesmosis in total syndesmotic ruptures.
Figure 2A. When I look at MRIs with isolated AITFL, in most cases I do not see an external rotation of the distal fibula.
Figure 2C. In these cases deltoid ligament injuries should also be excluded.
Line 454: Weber B fracture?
Line 457: Are special shaped titanium plates required for distal fibula fractures?
Line 489 ff: This discussion is rather a conclusion.
Line 498: How do you define working well? What about post traumatic arthritis? When is the typical onset?
According to the COI: M.R. had no role in the design of the study in the collection, analyses or interpretation of the data, in the writing of the manuscript or in the decision to publish the results.
However, he is listed as first author and when looking for the author contributions he was involved in everything. (Conceptualization, methodology, investigation, resources, data curation, writing, review and editing, supervision)
Reviewer 3 Report
Dear Authors,
I read and appreciated your paper, for the rigorous methodology, the clarity of expression, the relevance of the chosen topic.
I find the manuscript valid and useful for many colleagues.
I suggest to "stress" a little more the concept, as already you expressed, that the anatomical reduction is crucial in preventing the posttraumatic osteoarthritis, and above all how posttraumatic osteoarthritis has a large and negative impact on the patients quality of life (you can mention D’ambrosi, et al (2019). Post-traumatic ankle osteoarthritis: Quality of life, frequency and associated factors. Muscles, Ligaments and Tendons Journal, 9(3), 363-371. https://doi.org/10.32098/mltj.03.2019.10)
Round 2
Reviewer 2 Report
Thanks for the revision. I still believe that you the review is too broad and needs to focus on one type of injury especially when submitting it in the sports medicine section of the JCM.
For conflict of interest you mention the tightrope/ internal brace several times, however similar devices from other companies exist. Therefore, it should be rather called endobutton suture device.
In addition I may reference following statement: "it showed that patients treated with the suture-button device spent on average $1482" Zhang et al: A systematic review of suture-button versus syndesmotic screw in the treatment of distal tibiofibular syndesmosis injury published in BMC Musculoskeletal disorders (2017)
I am pleased for the explanations in the cover letter however only few changes have been made in the manuscript. I.e. Line 120, 130, 158, 208ff have not been addressed in the manuscript.
Typically gold standards are based on profound originally studies which highlight significant differences between two techniques allowing to develop an algorithm. In this case it is rather an expert opinion not to use them anymore.
Line 388: Thanks for the explanation. All three TFLs are generally accepted for a long time.
I
Author Response
Revision 2
Dear reviewer,
thank you very much again for your thorough review.
Thanks for the revision. I still believe that you the review is too broad and needs to focus on one type of injury especially when submitting it in the sports medicine section of the JCM.
We really tried to come up with all your suggestions, which clearly have helped to improve our manuscript so far. And we are very well aware of the fact that our manuscript incorporates different aspects of unstable syndesmotic injuries, which all would be worth to be dealt with in a separate manuscript, but unfortunately these different aspects often come together in real cases which the orthopaedic or trauma surgeon has to deal with in one single operation. Therefore, our very special intention was to give the treating physician a complete evidence-based treatment guideline from the first to the last aspect of the surgical treatment of unstable syndesmotic injuries. Our main aim was to provide an up-to-date and practicable solution for the question: “How should I currently treat unstable syndesmotic injuries according to the best available evidence in literature?” Neither more nor less. Moreover, most of the studies we have cited did not focus on a special singular injury pattern as well, and from a surgical point of view, it does not really make a great difference if you treat syndesmotic injuries as isolated injuries or combined with fractures: in the case of an unstable syndesmotic injury combined with fractures you usually stabilize the bony injuries first, thus transforming the combined syndesmotic injury into an isolated syndesmotic injury, and the final treatment of the remaining syndesmotic instability is quite the same.
We really think that splitting our manuscript into different parts would destroy our main intention described above. Therefore, we are unable to fulfill this special aspect of your request. We really hope that you can understand our argumentation, as we think that our manuscript would really be of benefit to orthopaedic and trauma surgeons treating unstable syndesmotic injuries.
For conflict of interest you mention the tightrope/ internal brace several times, however similar devices from other companies exist. Therefore, it should be rather called endobutton suture device.
Thank you very much for this valuable note. However, EndobuttonTM is also a protected trade mark held by Smith + Nephew, so the term you suggested is also not a really neutral expression. Therefore, we changed the term “tightrope” into the term “suture-button-device” where appropriate. However, one should be aware of the fact that many surgeons in the whole world are understanding the term “tightrope” much better than the term “suture-button-device”, especially in the context of syndesmotic injuries. The term “tightrope” has already become quite a well-known synonym for the term “suture-button-device”, and most of the published studies dealing with suture-button-devices for syndesmotic injures indeed were conducted using the TightRope® from Arthrex. Several cited references have even included the term “tightrope” in the title, and reporting results of these studies we had to use the correct term as well of course (please see references 120 and 124, for example). And moreover, as you mentioned the conflict of interest, none of the authors receives royalties due to the TightRope® implant system, and the two authors named in the conflict of interest (M.R. and G.M.) receive royalties from Arthrex due to other orthopaedic implant systems.
However, to avoid any inappropriate relations to the company Arthrex, we changed the protected term “TightRope®” into the term “tightrope” where possible and appropriate (please see lines 85, 87, 248, and 437), and we have added a remark that alternative suture-button-devices provided by other companies than Arthrex are available (please see lines 474ff).
In addition I may reference following statement: "it showed that patients treated with the suture-button device spent on average $1482" Zhang et al: A systematic review of suture-button versus syndesmotic screw in the treatment of distal tibiofibular syndesmosis injury published in BMC Musculoskeletal disorders (2017)
The authors are really aware of the fact that suture-button-devices are expensive compared to syndesmotic screws, and we had already clearly stated this fact in our discussion. However, definitive prices for such implants show a great international variability. For example, the current price for a tightrope system in Germany is about € 370, and also within one country prices usually vary due to special discounts the one or other institution gets from the manufacturer. And besides we had already cited the reference you have mentioned (please see reference 171).
I am pleased for the explanations in the cover letter however only few changes have been made in the manuscript. I.e. Line 120, 130, 158, 208ff have not been addressed in the manuscript.
We changed the manuscript accordingly and included the additional information as requested. Please see lines 127ff, 140ff, 161ff, and 222ff.
Typically gold standards are based on profound originally studies which highlight significant differences between two techniques allowing to develop an algorithm. In this case it is rather an expert opinion not to use them anymore.
As we have shown in our manuscript, there are at least 7 randomized controlled studies which are highlighting significant differences between suture-button-devices and syndesmotic screws in clear favor of the suture-button-devices. So, we really think this is high-quality scientific evidence and not only our expert opinion. Our conclusion that syndesmotic screws should no longer be considered the gold standard is based on a great number of profound original studies. And not to be misunderstood: it was currently not our aim to define a new gold standard, and in this regard, you would be absolutely right that we would first need some high-quality clinical studies to show that our new algorithm we have introduced is working.
However, according to your concerns, we changed our manuscript and chose a more reserved expression concerning the gold standard, please see lines 222 ff.
Line 388: Thanks for the explanation. All three TFLs are generally accepted for a long time.
We agree.